# Optimal free-surface pumping by an undulating carpet

Anupam Pandey [1], Zih-Yin Chen[2], Jisoo Yuk[3], Yuming Sun[4], Chris Roh[3], Daisuke Takagi[5], Sungyon Lee[2] & Sunghwan Jung [3]

Examples of fluid flows driven by undulating boundaries are found in nature across many different length scales. Even though different driving mechanisms have evolved in distinct environments, they perform essentially the same function: directional transport of liquid. Nature-inspired strategies have been adopted in engineered devices to manipulate and direct flow. Here, we demonstrate how an undulating boundary generates large-scale pumping of a thin liquid near the liquid-air interface. Two dimensional traveling waves on the undulator, a canonical strategy to transport fluid at low Reynolds numbers, surprisingly lead to flow rates that depend non-monotonically on the wave speed. Through an asymptotic analysis of the thin-film equations that account for gravity and surface tension, we predict the observed optimal speed that maximizes pumping. Our findings reveal how proximity to free surfaces, which ensure lower energy dissipation, can be leveraged to achieve directional transport of liquids.

The necessity to manipulate flow and transport liquids is primitive to many biophysical processes such as embryonic growth and development[1,2], mucus transport in bronchial tree[3–5], the motion of food within intestine[6,7], animal drinking[8,9]. Engineered systems also rely on efficient liquid transport such as in heat sinks and exchangers for integrated circuits[10,11], micropumps[12,13], and lab-on-a-chip devices[14]. Transporting liquids at small scales requires non-reciprocal motion to overcome the time reversibility of low Reynolds number flows. Deformable boundaries in the form of rhythmic undulation of cilia beds and peristaltic waves are nature's resolutions to overrule this reversibility and achieve directional liquid transport. While peristaltic pumps have become an integral component of biomedical devices, artificial ciliary metasurfaces that can actuate, pump, and mix flow have been realized only recently[15–19].

The design strategy of valveless micropumps essentially relies on a similar working principle as cilia-lined walls; sequential actuation of a channel wall by electrical or magnetic fields creates a traveling wave that drags the liquid along with it[20,21]. While the primary focus of micropumps has been on the transport of liquids enclosed within a channel, numerous technological applications require handling liquids near fluid–fluid interfaces. In particular, processes such as self-assembly, encapsulation, and emulsification involving micron-sized particles critically rely on the liquid flow near interfaces[22,23]. Liquid–air interfaces also act as a cradle to Neuston, organisms that inhabit at and below the water surface. For example, the underwater apple snail *Pomacea canaliculata* exploits the water surface to drive a large-scale surface flow and fetch floating food particles from afar in a process called pedal surface collection[24–27]. Whirligig beetles and water striders have developed morphological features to manipulate flow around them for effective locomotion at the interface[28,29]. Understanding the physics behind these natural phenomena could open up new bio-inspired strategies for flow actuation and sensing at interfaces.

Here, we reveal how a rhythmically deforming solid boundary pumps viscous liquid at the interface, and transports floating objects from distances much larger than its size. Our design, inspired in part by the capability of underwater snails in creating flow through undulations on their flexible foot[26,27], produces traveling waves on an undulator. Even though traveling boundaries are a canonical strategy to

[1]Mechanical & Aerospace Engineering Department and BioInspired Syracuse, Syracuse University, Syracuse, NY 13244, USA. [2]Department of Mechanical Engineering, University of Minnesota, Minneapolis, MN 55455, USA. [3]Department of Biological & Environmental Engineering, Cornell University, Ithaca, NY 14853, USA. [4]Sibley School of Mechanical & Aerospace Engineering, Cornell University, Ithaca, NY 14853, USA. [5]Department of Mathematics, University of Hawaii at Manoa, Honolulu, HI 96822, USA. ✉e-mail: apande05@syr.edu; sj737@cornell.edu

drive flow within enclosed spaces when placed near a liquid–air interface, the undulator gives rise to non-intuitive observations; pumping does not increase proportionally to the wave speed, and we observe non-monotonicity in the average motion of surface floaters as the wave speed is gradually increased. Detailed measurements of the velocity field in combination with an analysis of the lubrication theory unravel the interfacial hydrodynamics of the problem that emerges from a coupling between capillary, gravity, and viscous forces. We find that the non-monotonic flow is a direct consequence of whether the interface remains flat or conforms to the phase of the undulator. Through the theoretical analysis, we are able to predict the optimal wave speed that maximizes pumping, and this prediction is in excellent agreement with experiments. Finally, we show how pumping near an interface is a less dissipative strategy to transport liquid compared to pumping near a rigid boundary.

## Results

### Experiments

A 3D-printed undulator capable of generating traveling waves is attached to the bottom of an acrylic tank. The tank is filled with a viscous liquid (silicone oil or glycerin–water mixture) such that the mean depth of liquid above the undulator ($H$) remains much smaller the undulator wavelength ($\lambda$), i.e., $H/\lambda \ll 1$. The undulator is driven by a servo motor attached to a DC power source. Millimetric styrofoam spheres are sprinkled on the liquid surface and their motion is tracked during the experiment to estimate the large-scale flow of liquid. Additionally, we characterize the flow within the thin film of liquid directly in contact with the undulator by performing 2D particle image velocimetry (PIV) measurements. Our experimental design is essentially a mesoscale realization of the Taylor's sheet[30] placed near a free surface[31,32]; the crucial difference, however, is that the sheet or undulator is held stationary here, in contrast to free swimming.

Images of the undulator are shown in Fig. 1a, b. The primary component of this design is a helical spine encased with a series of hollow, rectangular links that are interconnected through a thin top surface[33] (see SI and Supplementary Movies 1 and 2 for details). The links along with the top surface form an outer shell that transforms the

helix rotation into a planar traveling wave of the form, $\delta \sin[2\pi(x - V_w t)/\lambda]$. The pitch and radius of the helix determine the wavelength $\lambda$ and amplitude $\delta$ of the undulations respectively. By modulating the angular frequency of the helix, we are able to vary the wave speed $V_w = \omega\lambda$ from 15 to 120 mm/s ($\lambda$ is fixed at 50 mm). We perform experiments with undulators of length $\lambda$ and $2\lambda$, and the results remain invariant of the undulator size. For given $V_w$, shapes of the undulator surface are shown in Fig. 1c for one period of oscillation.

**Large-scale flow.** Figure 1d shows the trajectories of floating styrofoam particles generated by 30 min of continuous oscillations in 1000 cSt silicone oil contained in an acrylic tank of dimensions $61 \times 46$ cm (Supplementary Movie 3 shows the motion of surface floaters for different $V_w$). Traveling waves on the actuator move in the downward direction as shown by the direction of $V_w$ in Fig. 1d. The large-scale flow caused by these undulations drags the floaters. The color code on the trajectories represents the arrow of time: blue and yellow colors represent the initial and final positions, respectively. Placing the undulator near a side wall of the tank, we measure the floaters' motion over a decade in distance. We quantify pumping from the net flow that the undulator creates near the free surface. Unlike closed pipes and channels, this net flow is not driven by an imposed pressure gradient. The shape of the free surface, which is an unknown a priori, determines the local pressure inside the liquid film. The undulator's open ends and proximity to the tank walls result in flow recirculation observed in Fig. 1d.

Typical Stokes number, $St = \rho_p R_p V_w/\eta$ remains very small, $\simeq 10^{-2}$ for the styrofoam floaters (based on wave speed $V_w = 100$ mm/s, particle radius $R_p = 1$ mm, particle density $\rho_p \simeq 50$ kg/m$^3$ and silicone oil viscosity $\eta_s = 0.97$ Pa · s). Thus we utilize the trajectories of these particles to estimate fluid motion at the interface. The particle tracks in Fig. 1d reveal that some particles are recirculated back in the flow due to the nearby tank wall, which we ignore in the analysis, and only focus on the floaters whose initial positions are straight ahead of the undulator. These trajectories are shown in Fig. 1e with black circles representing the initial positions. For a given $V_w$, we interpolate 20 trajectories to construct a velocity-distance curve which is shown in

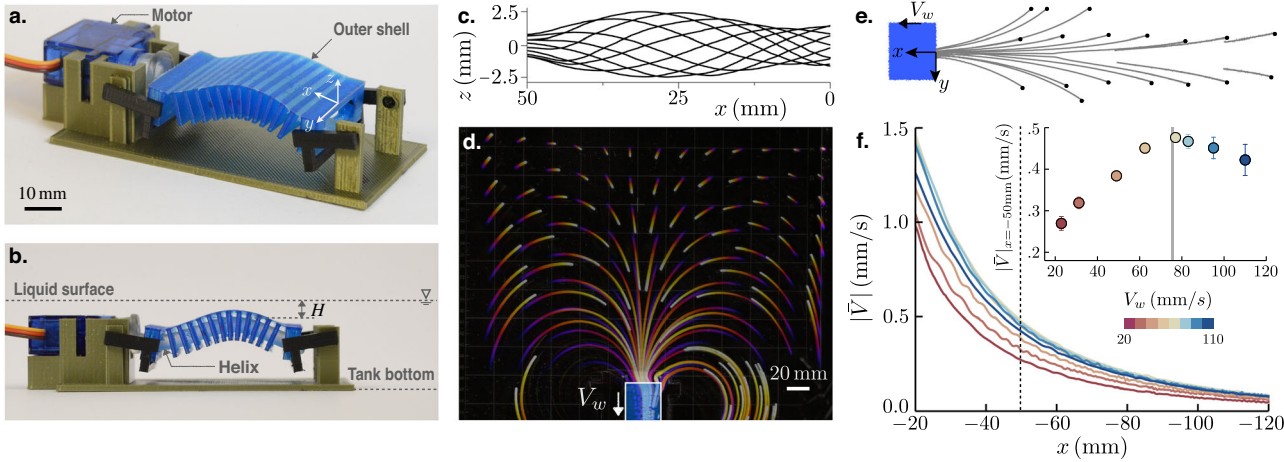

**Fig. 1 | Large-scale transport of floaters by the undulating carpet.** The actuator, shown in panels **a** and **b**, is comprised of a helix rotating inside a blue shell. Rotation of the helix causes an oscillatory motion of the shell forming a traveling wave on the surface. It is placed at a mean depth $H$ below the liquid surface. **c** Shape of the undulations over a period of oscillation. These shapes are captured by a traveling sine wave of $\delta \sin[2\pi(x - V_w t)/\lambda]$. **d** Trajectories representing motion of styrofoam particles at the interface due to 30 min of continuous oscillation of the undulator in silicone oil (viscosity 0.97 Pa · s) at a constant $V_w$. This panel is a top-view image with the actuator position marked at the bottom of the frame. The color coding of dark to light indicates the arrow of time. **e** Magnified trajectories of particles located

straight ahead of the actuator. The filled circles represent the initial positions of the styrofoam particles. **f** Particle velocity as a function of distance for increasing wave speeds ($V_w$). Different wave speeds are marked by the color coding. Distances are measured from the edge of the actuator, as shown in panel **e**. Each of the curves is an average of over 20 trajectories. Particle velocity exhibits a non-monotonic behavior with $V_w$ with maximum velocities measured at intermediate wave speeds. The inset confirms this behavior by showing particle velocity at a fixed location, $x = 50$ mm for different $V_w$. Error bars in this plot represent standard deviation in velocity magnitude. The gray line is the prediction from eq. (8).

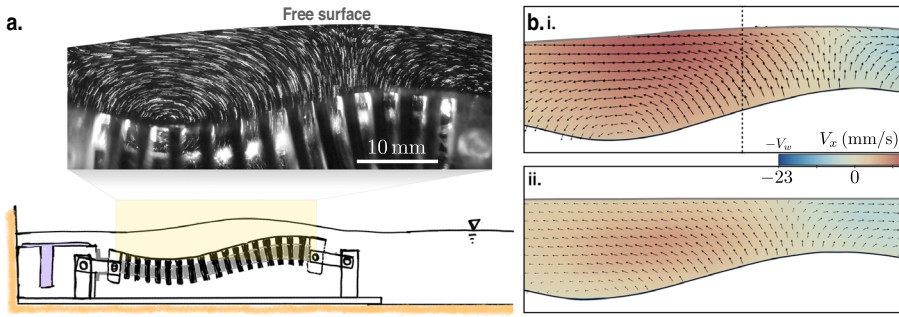

**Fig. 2 | Thin-film flow atop the undulator. a** A sketch of the actuator and a long-exposure image of a typical flow-field measurement, showing the motion of the tracer particles in the thin film. The free surface deforms in response to the flow. **b** Results of PIV for two different capillary numbers, $Ca = 714$ (top panel), $Ca = 17$ (bottom panel). In both these panels the bottom boundary is the actuator surface, while the top boundary is the liquid interface. The color coding represents the horizontal component of the velocity field, $V_x$; red signifies flow along the wave speed ($V_w$) while blue signifies flow opposite to $V_w$.

Fig. 1f (see SI for details of these measurements). Here, $|\bar{V}| = (V_x^2 + V_y^2)^{1/2}$ is the magnitude of the velocity at the liquid–air interface, and $x$ is the distance from the edge of the actuator. At the undulator edge, a sharp change in the boundary gives rise to a transition region in the flow field where surface floaters exhibit non-uniform and unsteady motion (cf. Fig. S2b in SI). Thus, we disregard the first 20 mm of data to avoid edge effects. The color code on the curves in Fig. 1f represents the magnitude of $V_w$. Interestingly, we observe a non-monotonic response in the surface fluid velocity; for any distance ($-20 \leq x \leq 120$) speed of fluid parcels initially increases with $V_w$, but subsequently drops down with a further increase in $V_w$. Once $|\bar{V}|$ at a fixed location ($x = -50$ mm) is plotted against $V_w$ (inset of Fig. 1f), it becomes apparent that the maximum surface flow is achieved for an intermediate speed, $V_w \simeq 80$ mm/s. Since the overall flow in the liquid is driven by the hydrodynamics within the thin film of liquid atop the undulator, we focus on quantifying the velocity field and flow rate in this region.

**Dimensionless groups.** Before we discuss the experimental results further, it is instructive to identify the relevant dimensionless groups that dictate the response of the system. The system has eight-dimensional parameters: three length scales given by film thickness $H$, amplitude ($\delta$) and wavelength ($\lambda$) of the undulator, the velocity scale $V_w$, gravitational constant $g$, and three fluid properties set by surface tension ($\gamma$), density ($\rho$), dynamic viscosity ($\eta$). These parameters lead to 5 dimensionless groups, namely, $\epsilon = \delta/H, a = H/\lambda$, Reynolds number $Re = \rho V_w \lambda a^2/\eta$, Capillary number $Ca = \eta V_w/(\gamma a^3)$, and Bond number $Bo = \rho g \lambda^2/\gamma$. Here both $Re$ and $Ca$ are defined for the thin-film limit, $a \ll 1$. We choose two working liquids, silicone oil ($\eta_s = 0.97$ Pa·s, $\gamma_s = 0.021$ N/m) and glycerin–water mixture (85% glycerin and 15% water by volume), GW ($\eta_{GW} = 0.133$ Pa·s, $\gamma_{GW} = 0.067$ N/m). For each of the liquids, the thickness, $H$ (maintaining $a \ll 1$), and wave speed, $V_w$ are varied independently. Across all experiments, $Re$ remains lower than 1 ($0.01 - 0.45$). Thus inertial effects are subdominant and the problem is fully described by $\epsilon, Ca$, and $Bo$. We vary $Ca$, the ratio of viscous to capillary forces, over three orders in magnitude, $7 - 6049$. The value of $Bo$, representing the strength of gravitational forces to surface tension, is 1133 and 426 for silicone oil and glycerin–water mixture respectively. For a full list of experimental parameters and dimensionless numbers, see Table S1 in SI. As we will demonstrate in the next sections, $Ca/Bo$, which represents the ratio of viscous force to gravitational force, turns out to be the key governing parameter.

**Non-monotonic flow rate.** The flow field within the thin film of liquid above the undulator is characterized by performing PIV at five longitudinal planes along the width of the undulator (see "Methods" section and SI section III for details). We find that the velocity field remains invariant of the PIV plane and exhibits low magnitudes of divergence, signifying a dominant 2D flow in the thin film. Figure 2a shows a long-exposure image of illuminated tracer particles, giving a qualitative picture of the flow. The particles essentially oscillate up-down with the actuator, but exhibit net horizontal displacement over a period due to the traveling wave. The presence of an interface is also crucial to the transport mechanism; the interfacial curvature induces a capillary pressure that modifies the local flow field. The coupling between the two deforming boundaries determines the flow within the gap. Snapshots of typical velocity fields for the two liquids are shown in Fig. 2b. The top panel is a silicone oil flow-field with $V_w = 23$ mm/s and $H = 5.7$ mm ($Ca = 714, Bo = 1133$), while the bottom panel represents flow-field of glycerin–water mixture with $V_w = 17$ mm/s and $H = 6.3$ mm ($Ca = 17, Bo = 426$). For both liquids, the instantaneous profile of $V_x$ remains half-parabolic along the depth of the thin film, which signifies a shear-free liquid–air interface. Higher $Ca$ leads to larger deformation of the free surface. Colors in the plot represent the magnitude of the horizontal velocity component, $V_x$; a portion of the liquid that follows the wave is shown in red, whereas a blue region represents part of the liquid that moves in the opposite direction to the wave. In fact, the velocity vectors at a given location switch directions depending on the phase of the actuator (see Supplementary Movies 4 and 5). Thus, to estimate the net horizontal transport of liquid across a section, we first integrate $V_x$ across thickness in the middle of the undulator (marked by the black dashed line in Fig. 2b, i), which yields an instantaneous flow rate

$$Q = \int_{h_a}^{h_f} V_x \, dz. \tag{1}$$

Here, $h_a$ and $h_f$ are the positions of the bottom and top boundaries from the reference point, respectively. Figure 3a plots $Q$ as a function of time, measured in silicone oil for three distinct wave speeds. It shows that $Q$ oscillates with the same time period as the undulator ($\tau = \lambda/V_w$), but there is a net flow of liquid along the traveling wave. Thus a time-averaged flow rate,

$$\langle Q \rangle = \frac{1}{\tau} \int_0^{\tau} Q \, dt, \tag{2}$$

gives a measure of liquid transport by the undulator.

Figure 3b gives a comprehensive picture of the flow rate measured across all the experiments. $\langle Q \rangle$ is plotted against the characteristic flow rate $V_w H$. The geometric prefactor of $\epsilon^2$ is a direct consequence of the thin geometry of the flow[34]. Two interesting observations are in order; (i) regardless of the fluid properties, the flow rates at first increase linearly with $\epsilon^2 V_w H$, (ii) all the data sets other than the GW exhibit a non-monotonic behavior with flow rates reaching maximum values at intermediate $\epsilon^2 V_w H$. Note that $Re$ in GW experiments tends to reach $\mathcal{O}(1)$ at the higher speeds. Thus, it is

expected to behave differently from the silicone oil experiments where $Re \le 10^{-1}$. Additional data presented in Fig. S5 of SI for experiments in GW (performed at higher $\epsilon^2 V_w H$ than what is shown in Fig. 3b) and pure water show that inertial effects lead to a continuous increase of $\langle Q \rangle$ with wave speeds, and confirm that the non-monotonic flow rates are a feature of the low $Re$ regime. Importantly, the non-monotonic surface flow observed in Fig. 1f seems to be directly correlated to the flow within the thin film above the undulator. We would like to point out that these measurements remain invariant of the undulator size as shown in the SI (cf. Fig. S4) where we compare the time-averaged flow rates measured in single and double-wave undulators. In the next section, we develop a theoretical model to explain how the geometrical and material parameters combine to give the optimal wave speed that maximizes the flow rate.

## Theoretical framework

**Thin-film equation.** We consider the two-dimensional geometry depicted in Fig. 4a for the theoretical model. An infinite train of periodic undulations of the form $h_a = \delta \sin 2\pi(x - V_w t)/\lambda$ propagates on the actuator located at a mean depth of $H$ from the free surface. We analyze the flow in the thin-film limit, such that $a = H/\lambda \ll 1$. A key aspect of the problem is that the shape of the interface, $h_f$ is unknown along with the flow-field. The explicit time dependence in this problem is a direct manifestation of the traveling wave on the boundary. Thus in a coordinate system $(X, Z)$ moving with the wave, the flow becomes steady. A simple Galilean transformation relates these coordinates to the laboratory coordinates $(x, z)$: $X = x - V_w t$, and $Z = z$. Thus, we first solve the problem in the wave frame and then transform the solution to the lab frame. Leaving the details of the derivation in Materials & Methods, here we present the key results.

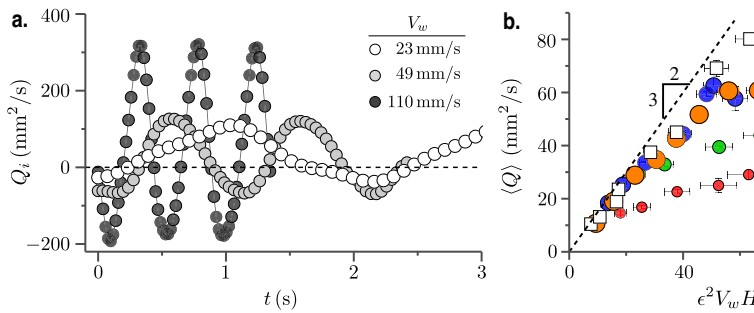

**Fig. 3 | Non-monotonic flow rate. a** Instantaneous flow rate in silicone oil over multiple periods of oscillation. The data sets represent increasing $V_w$, from white to black. These measurements are taken at a cross-section marked by the dashed line in Fig. 2b, i. **b** Time-averaged flow rate, $\langle Q \rangle$ is plotted against the flux scale $\epsilon^2 V_w H$ of the problem. The circles represent the experiment in silicone oil with bigger markers denoting larger height, $H$: 4.3 mm (red), 5.7 mm (green), 6.8 mm (blue), 8 mm (orange). The squares represent glycerin-water (GW) experiments with $H = 6.3$ mm. Error bars are based on standard deviation. The dashed line is the theoretical prediction, given by $\langle Q \rangle = 3\epsilon^2 V_w H/2$.

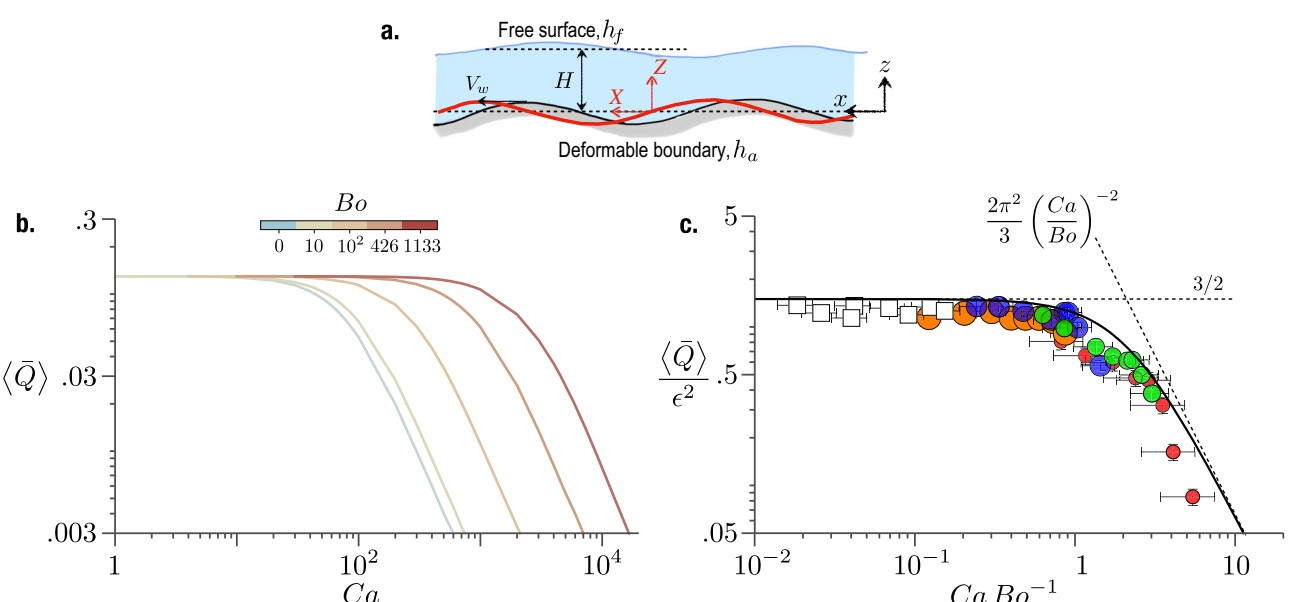

**Fig. 4 | Theoretical & numerical solutions of thin-film flow. a** The thin-film geometry with relevant quantities. We consider an infinite train of traveling undulations of amplitude $\delta$ and wavelength $\lambda$ moving at a speed of $V_w$. The coordinate frame $(X, Z)$ travels with the undulations. The red curve represents the bottom boundary in motion. A liquid layer of mean thickness $H$ reside on top the deformable bottom boundary. Shape of the free surface is given by $h_f$, while the bottom surface is given by $h_a$. **b** Numerical solution of (3) and (4), plotted in terms of the time-averaged flow rate $\langle \bar{Q} \rangle = \bar{q} + 1$ as a function of Capillary number ($Ca$), for different Bond numbers ($Bo$). The two large Bond numbers correspond to the experimental values. **c** The rescaled experimental data of Fig. 3b are in excellent agreement with the theoretical prediction of (7), plotted as the solid black line. Error bars are based on standard deviation. The small $V_w$ ($Ca/Bo \ll 1$) limit is given by $\langle \bar{Q} \rangle = 3\epsilon^2/2$, while the large $V_w$ ($Ca/Bo \gg 1$) limit is given by $\langle \bar{Q} \rangle = 2\pi^2 \epsilon^2 (Ca/Bo)^{-2}/3$.

In the thin-film limit, the separation of vertical and horizontal scales leads to a predominantly horizontal flow-field, and both mass and momentum conservation equations are integrated across the film thickness to reach an ordinary differential equation involving the free-surface shape, $h_f$, and volume flow rate $q$. Introducing dimensionless variables $\bar{X} = X/\lambda, \bar{h}_f = h_f/H, \bar{h}_a = h_a/H$, and $\bar{q} = q/V_w H$ we get

$$\bar{q} = \frac{1}{3}\left(\frac{1}{Ca}\bar{h}_f''' - \frac{Bo}{Ca}\bar{h}_f'\right)(\bar{h}_f - \bar{h}_a)^3 - (\bar{h}_f - \bar{h}_a), \qquad (3)$$

where both $\bar{q}$ and $\bar{h}_f$ are unknowns, and $\bar{h}_a = \epsilon \sin 2\pi \bar{X}$ is known. We close the problem by imposing the following constraint on $\bar{h}_f$:

$$\int_0^1 \bar{h}_f \, d\bar{X} = 1, \qquad (4)$$

which states that the mean film thickness over one wavelength does not change due to deformation. Along with periodic boundary conditions, equations (3) and (4) form a set of nonlinear coupled equations whose solutions depend on the three parameters, $Ca, Bo$, and $\epsilon$. For chosen $Bo$ and $\epsilon$, these equations are solved by a shooting method for a wide range of $Ca$. To be able to compare the numerical results with the experimental data of Fig. 3, we transform the results to the lab frame using the relation $\bar{Q} = \bar{q} + (\bar{h}_f(\bar{x},\bar{t}) - \bar{h}_a(\bar{x},\bar{t}))$. Owing to the periodic nature of $\bar{h}_f$ and $\bar{h}_a$, the time-averaged flow rate simplifies to $\langle \bar{Q} \rangle = \bar{q} + 1$. Figure 4b shows the numerical solution of $\langle \bar{Q} \rangle$ as a function of $Ca$ for $\epsilon = 0.3$ and different $Bo$. All curves exhibit the same qualitative behavior; at low $Ca$, the scaled flow rate reaches a constant value as $\langle Q \rangle \sim V_w H$, which is analogous to what we observe in Fig. 3b. At large $Ca$, however, we recover a decreasing flow rate as $\langle Q \rangle \sim (V_w H)^{-\alpha}$ with $\alpha > 0$. The transition between the two regimes scales with the $Bo$. Thus the thin-film equation captures the qualitative behavior found in the experiments. Interestingly, the non-monotonic response is also found in the $Bo = 0$ limit (light blue curve in Fig. 4b), broadening the significance of our results to smaller scales where capillary forces dominate over gravity.

**Asymptotic solution.** For $Bo \gg 1$, the third-order term in Eq. (3) can be neglected, which simplifies the governing equation to

$$\bar{q} = -\frac{1}{3}\frac{Bo}{Ca}\bar{h}_f'(\bar{h}_f - \bar{h}_a)^3 - (\bar{h}_f - \bar{h}_a). \qquad (5)$$

Indeed $Bo$ values in experiments are large (433 and 1132) justifying the above simplification. Furthermore, we assume that the amplitude of the wave, $\delta$ is much smaller than $H$, $\epsilon \ll 1$. Interestingly $Ca/Bo$, the single parameter dictating the solution of eq. (5), does not contain surface tension. This ratio is reciprocal to the Galileo number which plays a crucial role in the stability of thin films driven by gravity[35]. Here we look for asymptotic solutions of the form, $\bar{h}_f = 1 + \epsilon \bar{h}_{f1} + \epsilon^2 \bar{h}_{f2} + \mathcal{O}(\epsilon^3)$ and $\bar{q} = q_0 + \epsilon \bar{q}_1 + \epsilon^2 \bar{q}_2 + \mathcal{O}(\epsilon^3)$[36]. We insert these expansions in eqns. (5) and (4), and solve the equations in orders of $\epsilon$. Leaving the solution of $\bar{h}_f$ in the SI, here we present the solution of that becomes

$$\bar{q} = -1 + \frac{6\pi^2}{4\pi^2 + 9\left(Ca/Bo\right)^2}\epsilon^2. \qquad (6)$$

Thus the time-averaged flow rate in the lab frame is given by

$$\frac{\langle \bar{Q} \rangle}{\epsilon^2} = \frac{6\pi^2}{4\pi^2 + 9\left(Ca/Bo\right)^2}. \qquad (7)$$

This is the key result of the theoretical model. It demonstrates that the flow rate is quadratic in the amplitude of the traveling wave, which is why we incorporated $\epsilon^2$ in the horizontal scale of Fig. 3b.

Importantly, eq. (7) captures the non-monotonic behavior of the experiments. Once the data in Fig. 3b are rescaled, all the different cases collapse onto a master curve which is in excellent, overall agreement with the black solid line representing eq. (7), as shown in Fig. 4c. The difference between theory and experiment at large $Ca/Bo$ may be attributed to large deformations of the interface where the lubrication approximation of $\delta/H \ll 1$ breaks down.

**Optimal wave speed.** The physical picture behind the non-monotonic nature of the flow rate becomes clear once the free-surface shapes are found. For a given $Bo$ with a low $Ca$, the liquid–air interface behaves as an infinitely taut membrane with minimal deformations. Thus, a liquid parcel moves primarily in the horizontal direction, and the flow rate is given purely by the kinematics ($V_w, H, \delta$). Indeed for $Ca/Bo \ll 1$ eq. (7) simplifies to give $\langle \bar{Q} \rangle/\epsilon^2 = 3/2$. In the dimensional form, this relation explains the increase in the flow rate with the wave speed, $\langle Q \rangle = 3\delta^2 V_w/2H$. Thus, the flow rate is independent of the liquid properties which we have noted in Fig. 3b. As $Ca$ increases, the interface starts to deform up and down by conforming with the undulating actuator. In this limit, the translational velocity of tracer particles decreases thereby lowering the flow rate. Indeed, in the limit of $Ca/Bo \gg 1$, we find a decreasing flow rate given by $\langle \bar{Q} \rangle = 2\pi^2\epsilon^2(Ca/Bo)^{-2}/3$. These two asymptotic limits are shown as dashed lines in Fig. 4c. The flow rate attains a maximum at the intersection of these two lines where $Ca/Bo = 2\pi/3$. In the dimensional form, this particular value of $Ca/Bo$ gives the optimal wave speed at which flow rate peaks,

$$V_w^{(max)} = \left(\frac{2\pi\rho g H^3}{3\eta\lambda}\right). \qquad (8)$$

The optimal wave speed emerges from a competition between hydrostatic pressure ($\sim \rho g H$) and lubrication pressure ($\sim \eta V_w\lambda/H^2$); surface tension drops out in the above expression. Eq. (8) gives the optimal speed at which the undulator maximizes pumping. Now we are in a position to examine whether eq. (8) captures the peak surface velocities observed in Fig. 1f. Plugging in the density ($\rho = 970$ kg/m$^3$), viscosity ($\eta_s = 0.97$ Pa·s), $H = 5.7$ mm, $\lambda = 50$ mm, we find $V_w^{(max)} = 76.1$ mm/s, which matches very well with the observation (shown as the gray line in Fig. 1f inset). Note that the blue and orange data sets in Fig. 4c exhibit optimal flow rates at lower $Ca/Bo$ than the above asymptotic prediction. This difference could be attributed to higher $H/\lambda$ where the thin-film equation can not capture the exact flow-field.

**Pumping efficacy.** The flow rate achieved by this mechanism comes at the expense of the power needed to drive the undulator. This power expenditure equals the viscous dissipation within the flow. To this end, we estimate the efficacy of the mechanism by comparing the output, $\langle \bar{Q} \rangle$ to the input, viscous dissipation $\bar{\mathcal{E}}$ (see "Methods" for a derivation of $\bar{\mathcal{E}}$). To demonstrate the benefit of having an interface on the pumping capability of this mechanism, we compare it with the flux and dissipation for a rigid top boundary. These results are shown in Fig. 5. The data points represent dimensionless flux plotted against dissipation for a wide range of $Ca/Bo$. The $\epsilon \ll 1$ asymptotic result, shown as the black dashed line in Fig. 5, captures these results perfectly, giving the following algebraic relation between the two

$$\langle \bar{Q} \rangle = \bar{\mathcal{E}}. \qquad (9)$$

The importance of the free surface becomes apparent when the above result is compared to the scenario of the thin film bounded by a rigid, solid wall on top. As shown in the SI, for a rigid top boundary, both the flow rate and dissipation is given purely by the ratio of $\delta/H$. We find that the flow dissipates 4 times more energy to achieve the

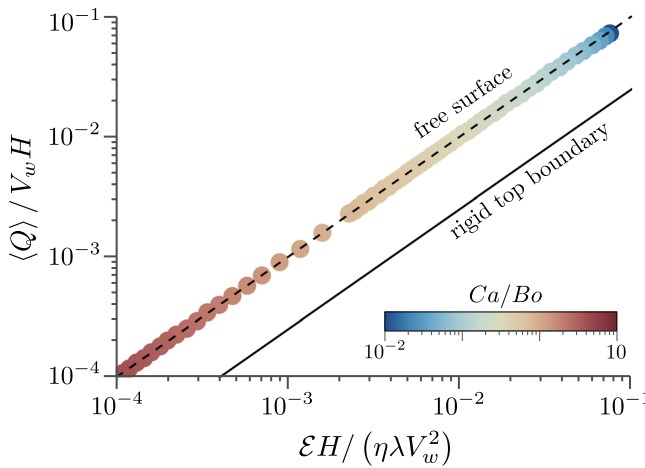

**Fig. 5 | Pumping efficacy of the undulator.** The dimensionless flow rate is plotted against the dimensionless dissipation for a wide range of *Ca/Bo* values. The data points represent numerical results, which are obtained for a fixed amplitude-to-depth ratio of $\epsilon = 0.3$. The dashed line is the asymptotic prediction of eq. (9). The solid line is the result of a top rigid boundary and represents eq. (10).

same amount of flow,

$$\langle \bar{Q} \rangle \simeq \frac{\bar{\mathscr{E}}}{4}. \tag{10}$$

This is plotted as the solid black line in Fig. 5. Thus it is clear that the liquid–air interface facilitates pumping by promoting horizontal transport of fluid parcels at a lower power consumption.

## Discussion

In summary, we have demonstrated the pumping capability of a sub-surface undulating carpet; the traveling wave triggers a large-scale flow beyond its body size. A direct observation of the liquid motion above the undulator in combination with a quantitative analysis of the thin-film equations, yields the optimal speed at which this device transports the maximum amount of liquid for given geometric and fluid properties. This optimal wave speed scales inversely with the wavelength of the undulations and linearly with the cube of the film thickness. It is interesting to note that the key governing parameter, *Ca/Bo* can be interpreted as a ratio of two velocities - wave speed ($V_w$) to a characteristic relaxation or leveling speed ($V_r = \rho g H^3/\eta \lambda$) at which surface undulations flatten out. This leveling process is dominated by gravity since the scale of undulations (~$\lambda$) is much larger than the capillary length ($\sqrt{\gamma/\rho g}$). Thus for $V_r \gg V_w$, the undulator essentially works against a relaxed, flat interface, and liquid parcels primarily exhibit horizontal displacement over a period. In the other limit of $V_r \ll V_w$, the free surface tends to beat in phase with the traveling boundary amplifying the vertical displacement, and subsequently reducing the net transport.

Our study demonstrates that the large-scale surface flow is a direct manifestation of the thin-film hydrodynamics above the undulator by showing how the optimal pumping speed captures the peak velocities in surface floaters. However, a quantitative analysis connecting the above two aspects of the flow field is necessary to exploit the full potential of this mechanism. Additionally, in the unexplored inertial regime, we expect the mechanism to showcase interesting dynamics due to the coupling between surface waves and finite-size particles[37]. In particular, for the transport of heavy or sedimentary particles, the Basset history force becomes important. This history force is proportional to the particle's size and density, especially for large or dense particles in an unsteady flow. Thus future studies on driven heavy particles and floating rafts could exhibit intricate, new

dynamics due to the Basset force. We believe that this work opens up new pathways for self-assembly and patterning at the mesoscale[38, 39], bio-inspired strategies for remote sensing and actuation within liquids[40,41], and control of interfacial flows using active boundaries[42,43].

## Methods

### Modeling & printing of the undulator
The models are designed in Fusion 360 (Autodesk). The helix is 3D printed in a Formlab Form 2 SLA printer by photo-crosslinking resin, whereas the outer shell comprising the top surface and rectangular links is printed in an Ultimaker S5 (Ultimaker Ltd.) using a blue TPE (thermoplastic elastomer). Due to the relative flexibility of TPE, the outer shell conforms to the helix. The helix is connected to a mini servo motor which is driven by DC power supply. All other parts (Base, Undulator holders, etc.) are printed using PLA (Polylactic acid) filaments on an Ultimaker S5 (Ultimaker Ltd.) printer.

### Measurement of the flow-field
We perform particle image velocimetry measurements on the thin liquid layer above the undulator. The viscous liquid is seeded with 10 μm glass microspheres (LaVision). A 520 nm 1W laser sheet (Laserland) illuminates a longitudinal plane in the middle of the undulator. Images are recorded by a Photron Fastcam SAZ camera at 500 frames per second. Image cross-correlation is performed in the open-source PIVlab[44] to construct the velocity field.

### Theoretical modeling
The separation of scales ($H \ll \lambda$) in the thin-film geometry leads to a set of reduced momentum equations and a flow field that is predominantly horizontal. Thus integration of the *X*-momentum equation with no-slip boundary condition on the undulator ($Z = h_a$) and no shear stress condition at the free surface ($Z = h_f$) results in

$$v_X = \frac{1}{2\eta} \frac{dp}{dX} \left[ (Z^2 - h_a^2) - 2h_f(Z - h_a) \right] - V_w. \tag{11}$$

Similarly, we integrate the *Z*-momentum equation and apply the Young-Laplace equation at the free surface, which yields the following expression for the pressure *p*,

$$p = -\gamma h_f'' + \rho g (h_f - Z). \tag{12}$$

Integration of the continuity equation gives the volume flow rate, $q = \int_{h_a}^{h_f} v_X dZ$, (per unit depth in this two-dimensional case). Plugging eqs. (11) and (12) into the expression of flow rate gives the following ODE,

$$q = \frac{1}{3\eta} (\gamma h_f''' - \rho g h_f')(h_f - h_a)^3 - V_w(h_f - h_a). \tag{13}$$

This equation relates the yet unknown constant *q* and the unknown free-surface shape $h_f$. We close the problem by imposing the following additional constraint on $h_f$:

$$\int_0^\lambda h_f \, dX = H\lambda, \tag{14}$$

which states that the mean film thickness over one wavelength remains the same as that of the unperturbed interface, *H*. In dimensionless form, the above set of equations take the form of eqs. (3) and (4) of the main text.

For a direct comparison with experiments, we transform the flow rate, *q* back to the lab frame which is an explicit function of time, $Q(x, t) = q + V_w(h_f(x, t) - h_a(x, t))$. We seek for periodic free-surface shapes, such that $h_f = H + $ periodic terms. The time-averaged flow

rate thus simplifies to

$$\langle Q \rangle = q + V_w H, \qquad (15)$$

where the integration is performed at a fixed spatial location. In dimensionless form, this equation becomes, $\langle \bar{Q} \rangle = \bar{q} + 1$, as mentioned in the main text.

To estimate the efficacy of the pumping mechanism, we compare the output, and flow rate to the energy dissipation within the flow, which is given by,

$$\mathcal{E} = \eta \int_{h_a}^{h_z} \int_0^\lambda \left( \frac{\partial v_X}{\partial Z} \right)^2 dX dZ. \qquad (16)$$

Using the expression for the velocity field of eq. (11), we integrate on $Z$ to find that the free-surface shape $h_f$ fully determines the amount of dissipation in the flow. In dimensionless form, dissipation becomes

$$\bar{\mathcal{E}} = \frac{1}{3} \left( \frac{Bo}{Ca} \right)^2 \int_0^1 \bar{h}_f'^2 \left( \bar{h}_f - \bar{h}_a \right)^3 d\bar{X}, \qquad (17)$$

where $\bar{\mathcal{E}} = \mathcal{E} H / (\eta \lambda V_w^2)$.

## Data availability
Source data are provided in this paper. Additionally, all data and Mathematica scripts are available on the Open Science Framework (DOI 10.17605/OSF.IO/ERZ79). Source data are provided in this paper.

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

## Acknowledgements

We thank Yohan Sequeira and Sarah MacGregor for their initial contributions. C.R., D.T., S.L., and S.J. acknowledge the support of NSF through grant no CMMI-2042740. A.P. acknowledges startup funding from Syracuse University.

## Author contributions

A.P., S.J. conceived the idea. A.P., J.Y., Y.S., C.R. and S.J. designed and performed experiments, and analyzed data. Z.C., D.T. and S.L. developed the theoretical and numerical models. All authors wrote the paper collectively.

## Competing interests

The authors declare no competing interests.
