## [Peer Review File · Nature Communications]

In the article “Optimal free-surface pumping by an undulating carpet” the authors have cleverly designed a “micropump” that generates travelling waves at the air/liquid interface that can be further employed to direct (interfacial) flow. The work is novel, the device is ingeniously designed and the article well-written and clear. However, the importance of this work to the general audience of Nature Communication seems to me quite limited. Similar setups, based for example on artificial cilia, have been continuously developed for the last ten years. The benefits of the different approach employed here, should in my opinion be clearly stated. Moreover, the theoretical model and specifically the assumption of completely mobile (or surface stress free) liquid/air interface should be critically discussed. The GW mixture should deviate from this assumption, as will almost all real-world systems and especially the biological ones.

Major comments:

1. The maximum in velocity is observed only for the silicone oil and not for the G/W mixture. However, no explanation is given by the authors. Why weren't the GW experiments continued to higher velocities (or thinner films) to see if a maximum is observed also for this system? Could this be related to Marangoni stresses?
2. What is the exact composition of the glycerine/water mixture? This is important as liquid mixtures and G/W specifically exhibit strong Marangoni stresses (see eg. the work of Lohse). Thus, the upper liquid/air surface might not be stress-free as assumed in the model of the authors.
3. The model employs the lubrication approximation (assuming that $H/\lambda \ll 1$). However, $\frac{H}{\lambda} \sim 0.2$. When will deviations from this assumption (e.g. due to u_z) show up? Similarly, it is assumed that $\delta/H \ll 1$, although it is $\frac{\delta}{H} \sim 0.35$. How will deviations from this assumption affect the predictions shown in Fig. 4c?
4. Although indeed $Re < 1$ inertial effects might not be negligible for high velocity for which $Re \sim 0.34$. Could this affect the occurrence of the non-monotonic behavior shown in Fig 1f?
5. In the physical sense, I don't understand how the solution of Eq. 5 can be independent of surface tension as the authors state. Why isn't the contribution of curvature i.e. $\frac{\sigma}{2r} \frac{\partial}{\partial r} \left(r \frac{\partial h_f}{\partial r} \right)$ included in the pressure balance of Eq. 12 given that $\frac{\delta}{H} \sim 0.35$? The occurrence of Marangoni stresses (and the dynamic surface tension or the surface tension gradients) which is probable for the model systems studied here, will also result in a dependence of flow speed on surface tension.
6. Figure 3b: The $\epsilon^2 V_w H$ at which the maximum in Q is observed does not show a gradual increase. For example the maximum for $H=7.5$ mm is observed at the same $\epsilon^2 V_w H$ as the $H=14$ mm and the $H=9.5$ mm in an even larger $\epsilon^2 V_w H$. Shouldn't V_w have actually a strong dependence on H (i.e. $V_w \sim H^3$) based on Eq.8?

Minor comments:

The viscosity of the silicone oil is sometimes stated to be 1000 cst (1 Pa*s) (Page 2) or 0.001 Pa*s (Page 3).

Section Dimensionless groups (page 3): it would be useful if the authors reported the value of the surface tension for both systems.

Page 6: the authors state that “the flow is independent of the liquid properties..”. Although this is true for surface tension according to the theoretical model (see my related concerns above), the flow still depends on the viscosity of the liquid and its density.

Reviewer #2 (Remarks to the Author):

The authors made a 3D-printed undulator capable of generating traveling waves, which was attached to the bottom of an acrylic tank. They revealed how a rhythmically undulating solid boundary pumps viscous liquid at the interface, and transported floating objects from distances much larger than their size. This structure and concept have some interesting points. They found pumping does not increase proportionally to the speed of the traveling wave, and non-monotonicity in the average motion of surface floaters as the wave speed is gradually increased. They also obtained its theoretical model.

Their experimental results and theoretical model are based on two dimensions. In 2D, their results and model look fine. However, this pump's flow is more complex and should be studied in a higher dimension. The performance of the pump needs to be characterized in 3D and the unit of the flow rates can be mm^3/s rather than mm^2/s . If full 3D characterization is difficult, 2D analysis on different planes seems necessary. And the limitation of 2D works needs to be described.

Reviewer #3 (Remarks to the Author):

In this paper, the authors study the pumping and transport of fluid in a thin film near a liquid-air interface generated by an undulating surface. The key result seems to be the emergence of non-monotonic dependence of the flow rate with the traveling wave velocity. The authors develop a lubrication theory to explain this observation and propose the mechanism as a new pumping device. In summary, the experiments are essentially a variant of peristaltic pumping near an interface. The work is overall interesting, the analysis complements the experiments suitably, and should be considered for publication.

However, I do not feel that in the present form of the manuscript, the study completely lives up to the initial claim of this as an alternative pumping device. There are a few concerns and questions along this line that needs to be answered and explored.

1. Authors should consider exploring how this device pumps in adverse pressure gradients. Ciliary beds, as mentioned in the motivation for the paper, are well known to pump fluid against opposing pressure gradients. It would be interesting to explore some of these characteristics.

2. As mentioned in the discussions, the authors propose this as a mesoscale pumping device. Given the size and velocity scales explored in the problem, the reason inertial effects remained low is because of the choice of a highly viscous fluid. I presume in mesoscale problems and for the transport of fluids with viscosities close to water, inertial effects will not be negligible. How will that alter the pumping characteristics?

3. Similarly, for a mesoscale device one needs to worry about finite Stokes number effects for particles being transported. I understand that a comprehensive analysis of this is beyond the present scope of the paper. But the authors should discuss what are the possible implications for transporting heavy/sedimenting particles or finite Stokes number particles for which history terms are important. When do they become important and what can be considered to account for this from a design perspective.

4. Can the authors discuss the challenges and possible adaptations of this mechanism to transport mucosal (or non-Newtonian) fluid near an interface? As seen in the snail example or ciliary beds.

Minor comments:

1. In the caption of Fig. 1(d), it would be helpful to explicitly mention that the tracked particles are on the interface of the fluid.

2. Below equation (2), there is a line: "Two interesting observations ...". It would be nice to explicitly mention the two observations as First and Secondly (or (i) and (ii))...

3. It seemed to me that the present study was very much inspired by the previous observations by several authors of this paper on freshwater snail feeding. One may argue that this is a simplified experimental realization of that system. Yet the example of the snail is only abruptly mentioned in the introduction and not discussed later on. The authors should reconsider their phrasing and highlight their previous work appropriately.

Reviewer #4 (Remarks to the Author):

I have read the work of Pandey et al. with great interest. Their manuscript is well writing and illustrated, and overall, very thorough. The quality of the results and the completeness of the work is in line with what one can expect from a publication in nature communications. The work also constitutes a significant improvement in our understanding of free surface boundary driven flows in the low Reynolds limit. As such, I would thus recommend to accept this manuscript with minimal revisions.

Here are a few questions I have, which I hope will help improving the quality of the manuscript.

In Fig1, I am not sure why it is necessary to remove the data from 0 to -20mm. We should be able to see it and then the authors might explain why this data has to be discarded.

How robust is the optimal vs. position. In other words, why taking -50mm. Likewise, the error bars are increasing as we move to higher speeds. Is there a reason why that might be the case.

In Fig2a, the interface is sharp, while the picture is a long exposure. This is odd, and I imagine that the picture has been cropped.

It might be best to show the all scene. It would be nice to compare the profiles of h_f obtained in the model to that found in experiment. Fig4 would be a good place to do so.

In any case, I was impressed by the ingenious nature of the work and the quality of the theoretical treatment. I want to reiterate that this work deserves publication in nature comm.

Reply to reviewer 1:

In the article “Optimal free-surface pumping by an undulating carpet” the authors have cleverly designed a “micropump” that generates travelling waves at the air/liquid interface that can be further employed to direct (interfacial) flow. The work is novel, the device is ingeniously designed and the article well-written and clear. However, the importance of this work to the general audience of Nature Communication seems to me quite limited. Similar setups, based for example on artificial cilia, have been continuously developed for the last ten years. The benefits of the different approach employed here, should in my opinion be clearly stated. Moreover, the theoretical model and specifically the assumption of completely mobile (or surface stress free) liquid/air interface should be critically discussed. The GW mixture should deviate from this assumption, as will almost all real-world systems and especially the biological ones.

Reply: We thank the reviewer for the positive assessment of our work, and suggestion to highlight the novelty and benefit of our approach over existing work on apparently analogous systems such as artificial cilia. Majority of recent works on the development of ciliated artificial surfaces, for example the references [15]-[19] of the main text, focuses on the precise actuation and control of individual cilia to drive directional fluid transport and mixing. However, it is not known how an actuated cilia carpet would behave near a liquid-liquid or a liquid-air interface. The novelty of our study is that it reveals new, fundamental flow-physics when a traveling wave boundary is placed near a fluid-fluid interface. Recent studies have pointed out that a similar mechanism could be at play behind the yet unclear feeding mechanism of underwater apple snails. Thus our design could open up new, bio-inspired design strategies for flow actuation and sensing at an interface. In the revised introduction we have spelled out these connections to biological examples, and pointed out the novelty of our system

Liquid-air interfaces also act a cradle to Neuston, organisms that inhabit at and below the water surface. For example, the underwater apple snail *Pomacea canaliculata* exploits the water surface to drive a large scale surface flow and fetch floating food particles from afar in a process called *pedal surface collection* [24-27]. Whirligig beetles and water striders have developed morphological features to manipulate flow around them for effective locomotion at the interface [28,29]. Understanding the physics behind these natural phenomena could open up new bio-inspired strategies for flow actuation and sensing at interfaces.

...Our design, inspired in part by the capability of underwater snails in creating flow through undulations of its flexible foot [26,27], produces travelling waves on an undulator. Even though travelling boundaries are a canonical strategy to drive flow within enclosed spaces, when placed near a liquid-air interface, the undulator gives rise to non-intuitive observations; pumping does not increase proportionally to the wave speed, and we observe non-monotonicity in the average motion of surface floaters as the wave speed is gradually increased.

We would like to point the reviewer to our response to question 2 where we address his/her concern on the assumption of a shear free interface for the GW experiments.

Major comments:

1. *The maximum in velocity is observed only for the silicone oil and not for the G/W mixture. However, no explanation is given by the authors. Why weren't the GW experiments continued to higher velocities (or thinner films) to see if a maximum is observed also for this system? Could this be related to Marangoni stresses?*

Reply: We thank the reviewer for bringing up this issue. This question is intricately connected to the question 4 by the reviewer. So here we reply both the questions. The GW data in fig. 3b of manuscript at higher speeds reach $Re = 0.45$ where inertial effects do not remain negligible. In fact, following the reviewer's suggestion, we have performed additional experiments in GW at speeds beyond what is shown in fig 3b, and new experiments in pure water. These data are shown in fig. S4 in SI and reproduced below. Distinct from the data in the main text, for $Re \sim O(1)$, the pumping rate $\langle \bar{Q} \rangle$ no longer exhibits the non-monotonic behavior and does not match the lubrication model that neglects the inertia. We are currently developing a new thin-film model that includes the effects of inertia to address this new physical regime, which remains outside the scope of this current manuscript

In the SI we have added the following description of the new experiment along with the figure

We supplement the data shown in fig. 3b of the main manuscript with additional experiments in glycerin-water mixture and water to confirm that the non-monotonic behavior in flow rates emerge only in the low Re regime. For GW, these additional experiments were performed at speeds (78.54 mm/s, 94.69 mm/s, and 108.07 mm/s) beyond what is presented in fig. 3b. For water we tested the wave speeds ranging from 28.21 mm/s to 117.46 mm/s. Measurements of time-averaged flux from these experiments are plotted in fig. S5. We include one set of silicone oil data in this plot for comparison. It is interesting to note that neither GW nor water exhibits a dip in the flux values at higher $\epsilon^2 V_w H$. We anticipate that the increasing role of inertia to be the underlying reason for the qualitatively different behavior in GW and water experiments. While $Re \leq 8 \times 10^{-2}$ for silicone oil, Re values in GW reach close to 1 (0.75 to be exact) at the highest speed. $Re \geq 1$ for all data points in water. Thus, a theoretical framework capable of handling finite Re effects is necessary to capture these experimental observations.

In the main manuscript, we have added

Note that Re in GW experiments tends to reach $O(1)$ at the higher speeds. Thus, it is expected to behave differently from the silicone oil experiments where $Re \leq 10^{-1}$. Additional data presented in fig. S5 of SI for experiments in GW (performed at higher $\epsilon^2 V_w H$ than what is shown in fig. 3b) and pure water show that inertial effects lead to

Figure 1: **Time averaged flow rate for water and GW mixture, showing monotonically increasing flux in contrast to silicone oil.** Gray squares represent the additional GW data at higher V_w where $Re \sim \mathcal{O}(1)$. $Re \geq 1$ for all the water experiments.

a continuous increase of $\langle Q \rangle$ with wave speeds, and confirm that the non-monotonic flow rates are a feature of the low Re regime.

2. *What is the exact composition of the glycerine/water mixture? This is important as liquid mixtures and G/W specifically exhibit strong Marangoni stresses (see eg. the work of Lohse). Thus, the upper liquid/air surface might not be stress-free as assumed in the model of the authors.*

Reply: Following the reviewer’s suggestion, we have specified the composition of GW mixture which is 85% glycerin and 15% water by volume.

We estimate the role of Marangoni velocity in GW as $\Delta\gamma/\eta$ which is of the order of 1 mm/s (for $\Delta\gamma \simeq 10^{-2}$ N/m and $\eta = .133$ Pa.s). This velocity is much smaller than V_w (varying between 20 to 120 mm/s) which is the velocity scale in the problem. Thus we anticipate that the role of Marangoni driven flow in the problem is sub-dominant.

The assumption of a stress-free interface in the theoretical model is based upon the velocity profiles measured within the thin-film through PIV. Few of those instantaneous velocity profiles in GW are shown in fig. 2. A half-parabolic nature of these profiles signifies that the liquid-air interface remain stress free. In the revised manuscript, we have spelled it out on page 4 -

For both the liquids, the instantaneous profile of V_x remains a half-parabolic along the depth of the thin film, which signifies a shear free liquid-air interface...

Figure 2: Velocity profiles are obtained through Particle Image Velocimetry (PIV) measurements. The free surface of the liquid is positioned approximately 5-6 mm along the z -axis, while the undulating boundary exhibits vertical and slight lateral movement. As a result, the starting points of the half-parabolic velocity profiles vary due to the undulator's motion. However, the majority of velocity profiles exhibit a consistent half-parabolic shape, which indicates the validity of the stress-free condition on the liquid-air interface.

3. The model employs the lubrication approximation (assuming that $H/\lambda \ll 1$). However, $H/\lambda \sim 0.2$. When will deviations from this assumption (e.g. due to u_z) show up? Similarly, it is assumed that $\delta/H \ll 1$, although it is $\delta/H \sim 0.35$. How will deviations from this assumption affect the predictions shown in Fig. 4c?

Reply: The referee is correct in pointing out that our lubrication parameter $a \equiv H/\lambda$ is not strictly small. However, a number of previous studies (e.g., [1,2,3,4]) have demonstrated that the lubrication approximation can work well beyond the strictly small-slope limits. Furthermore, the normal velocity component, $\overline{u_z}$, is negligible as long as $a^2 \ll 1$, the details of which are shown below.

The momentum and mass conservation equations can be expressed as

$$\frac{\partial \overline{u_x}}{\partial \overline{X}} + \frac{\partial \overline{u_z}}{\partial \overline{Z}} = 0, \quad (1)$$

$$a^2 \text{Re} \left(\overline{u_x} \frac{\partial \overline{u_x}}{\partial \overline{X}} + \overline{u_z} \frac{\partial \overline{u_x}}{\partial \overline{Z}} \right) = -\frac{\partial \overline{P}}{\partial \overline{X}} + a^2 \frac{\partial^2 \overline{u_x}}{\partial \overline{X}^2} + \frac{\partial^2 \overline{u_x}}{\partial \overline{Z}^2}, \quad (2)$$

$$a^4 \text{Re} \left(\overline{u_x} \frac{\partial \overline{u_z}}{\partial \overline{X}} + \overline{u_z} \frac{\partial \overline{u_z}}{\partial \overline{Z}} \right) = -\frac{\partial \overline{P}}{\partial \overline{Z}} + a^2 \left(a^2 \frac{\partial^2 \overline{u_z}}{\partial \overline{X}^2} + \frac{\partial^2 \overline{u_z}}{\partial \overline{Z}^2} \right) - \frac{\text{Bo}}{\text{Ca}}, \quad (3)$$

with Reynolds number $\text{Re} = \rho V_w \lambda / \mu$. From the above governing equations, we note that

$\overline{u_z}$ can be neglected as long as $a^2 Re \ll 1$ and $a^2 \ll 1$. Since the magnitude of $a^2 Re$ is around $O(10^{-2})$ to $O(10^{-1})$ in the experiments, we conclude that the effects of $\overline{u_z}$ can be neglected in the current study.

Finally, we also take the limit of $\delta/H \ll 1$ to further linearize the lubrication equation. The limit of $\delta/H \ll 1$ allows us to decompose the free surface shape \overline{h}_f into a constant value (i.e., $\overline{h}_f = 1$) at the leading order and the deviation from the flat interface at $O(\delta/H)$ or higher. Therefore, we expect that this assumption will break down as the free surface deforms significantly from the flat surface, which is in the limit of increasing Ca/Bo . Note that the effects of hydrostatic pressure tend to stabilize the free surface to become more flat, which coincides with the limit of $Ca/Bo \rightarrow 0$. Going back to Figure 4c, as Ca/Bo increases, we observe a growing deviation between the experimental data of $\langle \overline{Q} \rangle$ and the theoretical prediction. We speculate that this may be due to the breakdown of the asymptotic limit $\delta/H \ll 1$, as the free surface tends to deform more. In the revised manuscript we add the following comment on the deviation between the experiments and theory for large Ca/Bo .

The difference between theory and experiment at large Ca/Bo may be attributed to large deformations of the interface where the lubrication approximation of $\delta/H \ll 1$ breaks down.

4. Although indeed $Re < 1$ inertial effects might not be negligible for high velocity for which $Re \sim 0.34$. Could this affect the occurrence of the non-monotonic behavior shown in Fig 1f?

Reply: See our response to comment 1.

5. In the physical sense, I don't understand how the solution of Eq. 5 can be independent of surface tension as the authors state. Why isn't the contribution of curvature i.e. $\frac{\sigma}{2r} \frac{\partial}{\partial r} \left(r \frac{\partial h_f}{\partial r} \right)$ included in the pressure balance of Eq. 12 given that $\delta/H \sim 0.35$? The occurrence of Marangoni stresses (and the dynamic surface tension or the surface tension gradients) which is probable for the model systems studied here, will also result in a dependence of flow speed on surface tension.

Reply: We appreciate the referee's comment regarding surface tension. As the referee correctly pointed out, the effects of surface tension are indeed included in our model in the dynamic boundary condition (Eq. 12), which, in turn, shows up as the $Ca^{-1} \overline{h}_f'''$ term in the expression for the flux (Eq. 3). Note that (Eq. 3) also contains the effects of hydrostatic pressure in the form of $(Bo/Ca) \overline{h}_f'$. Based on the characteristic values in the experiments, Ca ranges from $O(1)$ to $O(10)$, while the magnitude of Bo is around $O(10^2)$ to $O(10^3)$. Therefore, in our current system, hydrostatic pressure dominates over surface tension, and it is reasonable to neglect the curvature term (i.e., $Ca^{-1} \overline{h}_f'''$) in (Eq. 3) to obtain an analytical solution, which eliminates surface tension from the model. Furthermore, even if we

retain the curvature term (i.e., $Ca^{-1}\overline{h}_f'''$) and solve (Eq. 3) numerically (see Fig. 4b), the solutions for \overline{h}_f and the resultant pumping rates $\langle Q \rangle$ do not strongly depend on surface tension for the range of Bo from the experiments, since $(Bo/Ca) \gg (1/Ca)$.

6. Figure 3b: The $\epsilon^2 V_w H$ at which the maximum in Q is observed does not show a gradual increase. For example the maximum for $H=7.5$ mm is observed at the same $\epsilon^2 V_w H$ as the $H=14$ mm and the $H=9.5$ mm in an even larger $\epsilon^2 V_w H$. Shouldn't V_w have actually a strong dependence on H (i.e. $V_w \sim H^3$) based on Eq.8?

Reply: We thank the reviewer for this insightful comment. The quantity $\epsilon^2 V_w H = \delta^2 V_w / H$ is in fact inversely proportional to H . Since the $V_w^{(max)}$ scales with H^3 (as per eq. 8), the value of $\epsilon^2 V_w H$ at which $\langle Q \rangle$ becomes maximum should increase with H^2 . This trend is apparent in the red and green data sets of fig. 3b, but the blue data does not. We think that the intermediate thickness corresponding to blue data falls in the transition regime (see the rescaled data of fig. 4c) where the asymptotic results do not hold.

Minor comments:

*The viscosity of the silicone oil is sometimes stated to be 1000 cst (1 Pa*s) (Page 2) or 0.001 Pa*s (Page 3).*

Reply: We thank the author for pointing this mistake out. We have corrected this discrepancy.

Section Dimensionless groups (page 3): it would be useful if the authors reported the value of the surface tension for both systems.

Reply: We have added the surface tension values in the section on ‘Dimensionless groups’.

Page 6: the authors state that “the flow is independent of the liquid properties..”. Although this is true for surface tension according to the theoretical model (see my related concerns above), the flow still depends on the viscosity of the liquid and its density.

Reply: With the above mentioned statement we intended to refer to the asymptotic result of $\langle Q \rangle = 3\delta^2 V_w / 2H$ in the low Ca/Bo limit. In this limit, the time averaged flow rate depends only on the geometric parameters, δ , H , and wave speed V_w . Thus the flow rate is independent of the fluid properties.

Reference

1. Chen, Z., Hooshanginejad, A., Kumar, S., & Lee, S. (2022). Droplet dynamics under an impinging air jet. *Journal of Fluid Mechanics*, 943, A32.
2. Hooshanginejad, A., Dutcher, C., Shelley, M., & Lee, S. (2020). Droplet breakup in a stagnation-point flow. *Journal of Fluid Mechanics*, 901, A19.
3. Espin, L., & Kumar, S. (2017). Droplet wetting transitions on inclined substrates in the presence of external shear and substrate permeability. *Physical Review Fluids*, 2, 014004.
4. Krechetnikov, R. (2010). On application of lubrication approximations to nonunidirectional flow.

tional coating flows with clean and surfactant interfaces. *Physics of Fluids*, 22, 092102.

Reply to reviewer 2:

The authors made a 3D-printed undulator capable of generating traveling waves, which was attached to the bottom of an acrylic tank. They revealed how a rhythmically undulating solid boundary pumps viscous liquid at the interface, and transported floating objects from distances much larger than their size. This structure and concept have some interesting points. They found pumping does not increase proportionally to the speed of the traveling wave, and non-monotonicity in the average motion of surface floaters as the wave speed is gradually increased. They also obtained its theoretical model.

Their experimental results and theoretical model are based on two dimensions. In 2D, their results and model look fine. However, this pump's flow is more complex and should be studied in a higher dimension. The performance of the pump needs to be characterized in 3D and the unit of the flow rates can be mm^3/s rather than mm^2/s . If full 3D characterization is difficult, 2D analysis on different planes seems necessary. And the limitation of 2D works needs to be described.

Reply: We thank the reviewer for pointing out the possible 3D effects in the flow-field. Indeed, the finite width of the undulator is responsible for a cross-stream velocity component. While a full quantification of this 3D flow-field is beyond the scope of the present work, following the suggestion of the reviewer, we have performed additional PIV measurements along 5 different longitudinal planes spaced 5 mm apart. The results are shown in the figure below which is added as fig. S3 in the SI. We find that the flow-field remains qualitatively similar across all the planes with low magnitude of divergence. This measurements confirm that barring the edges of the undulator, the 2D approximation works well. Instantaneous flow rate values are also found to be similar on the different planes. In the revised manuscript, we have added the following discussion and figure in the SI

We perform PIV at five longitudinal planes spaced across the width of the undulator, as shown in fig. S3 a. These planes are illuminated by a laser sheet. We find that the velocity field remains invariant across those planes signifying the 2D nature of the flow field within the thin film above the undulator. The instantaneous flow rate measured at the middle of the five planes remain identical over multiple oscillations as shown in fig S3 b. The qualitative similarity among the velocity fields on multiple planes are demonstrated in fig. S3 c along with a color map of the local divergence value. The divergence remains low except for the boundaries. These measurements corroborate our assumption of the predominantly two-dimensional nature of flow in the thin liquid layer...

In the 'Non-monotonic flow rate' section of the main manuscript, we have added the following argument -

The flow field within the thin film of liquid above the undulator is characterized by performing PIV at five longitudinal planes along the width of undulator (see Materials & Methods section and SI section III for details). We find that the velocity-field remains invariant of the PIV plane and exhibits low magnitudes of divergence, signifying a dominant 2D flow in the thin film...

Figure 3: **PIV at multiple planes across the width of the undulator.** a) PIV is performed on 5 longitudinal planes evenly spaced 5-mm apart. b) Instantaneous flow rate vs time as measured at the center of the five longitudinal planes in silicone oil. $V_w = 96$ mm/s and $H = 5.7$ mm for these measurements. c) Qualitative similarity among velocity fields on multiple longitudinal planes. The color map in the panels represents divergence of the flow-field which remain low in all the planes demonstrating predominantly 2D nature of the flow in the thin film.

Reply to reviewer 3:

In this paper, the authors study the pumping and transport of fluid in a thin film near a liquid-air interface generated by an undulating surface. The key result seems to be the emergence of non-monotonic dependence of the flow rate with the traveling wave velocity. The authors develop a lubrication theory to explain this observation and propose the mechanism as a new pumping device. In summary, the experiments are essentially a variant of peristaltic pumping near an interface. The work is overall interesting, the analysis complements the experiments suitably, and should be considered for publication.

However, I do not feel that in the present form of the manuscript, the study completely lives up to the initial claim of this as an alternative pumping device. There are a few concerns and questions along this line that needs to be answered and explored.

Reply: We thank the reviewer for the detailed comments and suggestions, and for the support of publication in Nature Communications.

1. Authors should consider exploring how this device pumps in adverse pressure gradients. Ciliary beds, as mentioned in the motivation for the paper, are well known to pump fluid against opposing pressure gradients. It would be interesting to explore some of these characteristics.

Reply: We thank the referee for this interesting suggestion. In the lubrication limit, an adverse pressure gradient across the length of the undulator is expected to give rise to a pumping rate that scales as $(h_f - h_a)^3$ in the opposite direction of the traveling wave, where $h_f - h_a$ is the thickness of the thin film. While we may reasonably assume that the adverse pressure gradient will reduce the total time-averaged pump rate, it is unclear how the combination of the adverse pressure gradient and the undulating surface deformations will modify the shape of the free surface, h_f . Hence, fully understanding the effects of the adverse pressure will require re-doing the lubrication analysis without the simplifying periodic boundary conditions. We believe this is outside the scope of our current manuscript but an interesting direction to consider for our future studies, both experimentally and mathematically.

2. As mentioned in the discussions, the authors propose this as a mesoscale pumping device. Given the size and velocity scales explored in the problem, the reason inertial effects remained low is because of the choice of a highly viscous fluid. I presume in mesoscale problems and for the transport of fluids with viscosities close to water, inertial effects will not be negligible. How will that alter the pumping characteristics?

Reply: We agree with the referee that the inertial regime is easily accessible in our experiments owing to the scale of our device. Following this suggestion, we have performed **additional experiments in water and glycerin-water mixture which are presented in fig. S5 (also attached below) of the SI**. These experiments are in the regime of $Re \sim O(1)$. Distinct from the data in the main text, for $Re \sim O(1)$, the pumping

Figure 4: **Time averaged flow rate for water and GW mixture, showing monotonically increasing flux in contrast to silicone oil.** Gray squares represent the additional GW data at higher V_w where $Re \sim \mathcal{O}(1)$. $Re \geq 1$ for all the water experiments.

rate $\langle \bar{Q} \rangle$ no longer exhibits the non-monotonic behavior and continues to increase with the wave speed, which does not match the results of our lubrication model with no inertia. This additional data strongly suggests that fluid inertia tends to enhance the net pump rate inside the thin film. However, it requires more systematic studies to fully understand the inertial effects on free surface pumping. To that end, we are currently developing a new thin-film model that includes the effects of inertia to address this new physical regime, which remains outside the scope of this current manuscript.

In the SI we have added the following description of the new experiment along with the figure

We supplement the data shown in fig. 3b of the main manuscript with additional experiments in glycerin-water mixture and water to confirm that the non-monotonic behavior in flow rates emerge only in the low Re regime. For GW, these additional experiments were performed at speeds (78.54 mm/s, 94.69 mm/s, and 108.07 mm/s) beyond what is presented in fig. 3b. For water we tested the wave speeds ranging from 28.21 mm/s to 117.46 mm/s. Measurements of time-averaged flux from these experiments are plotted in fig. S5. We include one set of silicone oil data in this plot for comparison. It is interesting to note that neither GW nor water exhibits a dip in the flux values at higher $\epsilon^2 V_w H$. We anticipate that the increasing role of inertia to be the underlying reason for the qualitatively different behavior in GW and water experiments. While $Re \leq 8 \times 10^{-2}$ for silicone oil, Re values in GW reach close to 1 (0.75 to be exact) at the highest speed. $Re \geq 1$ for all data points in water. Thus, a theoretical framework capable of handling finite Re effects is necessary to capture these experimental observations.

In the main manuscript, we have added

Note that Re in GW experiments tends to reach $\mathcal{O}(1)$ at the higher speeds. Thus, it is expected to behave differently from the silicone oil experiments where $Re \leq 10^{-1}$. Additional data presented in fig. S5 of SI for experiments in GW (performed at higher $\epsilon^2 V_w H$ than what is shown in fig. 3b) and pure water show that inertial effects lead to a continuous increase of $\langle Q \rangle$ with wave speeds, and confirm that the non-monotonic flow rates are a feature of the low Re regime.

3. Similarly, for a mesoscale device one needs to worry about finite Stokes number effects for particles being transported. I understand that a comprehensive analysis of this is beyond the present scope of the paper. But the authors should discuss what are the possible implications for transporting heavy/sedimenting particles or finite Stokes number particles for which history terms are important. When do they become important and what can be considered to account for this from a design perspective.

Reply: As the reviewer pointed out, the concern regarding finite Stokes number effects for particles being transported is indeed an important consideration. When considering heavy particles, the transport behavior can be significantly influenced by inertial effects, which include the Basset history force. They become important when the particles have a sufficiently large size or density in unsteady flow. In our study, the particle velocity fluctuated around its steady value, but the particle size and density were small. Hence, the Basset force is negligible.

In the discussion section of the revised manuscript, we add -

...In particular, for the transport of heavy or sedimentary particles, the Basset history force becomes important. This history force is proportional to the particle's size and density, especially for large or dense particles in an unsteady flow. Thus future studies on driven heavy particles and floating rafts could exhibit intricate, new dynamics due to the Basset force.

4. Can the authors discuss the challenges and possible adaptations of this mechanism to transport mucosal (or non-Newtonian) fluid near an interface? As seen in the snail example or ciliary beds.

Reply: We thank the referee for the insightful question. Yes, it is correct that snail's mucus is indeed non-Newtonian. In particular, the yield stress of the mucus is an essential features that allows the snails to move over a variety of terrains and provides a thrust force when they need to climb or crawl up steep inclines. However, the benefit of the non-Newtonian mucus in feeding is not clear yet. A major challenge in transporting mucus near an interface is that some regions may not flow or yield, which may require extra stresses to deform the material. Additionally, it is not practical to apply this technology directly onto bio-inspired robots due to the fact that any fluid applied will be washed away by flowing fluids and must then be continually resupplied if motion is desired over an extended period of time.

Minor comments:

1. *In the caption of Fig. 1(d), it would be helpful to explicitly mention that the tracked particles are on the interface of the fluid.*

Reply: Following the reviewer’s suggestion caption of fig. 1d has been modified to -

Trajectories representing motion of styrofoam particles at the interface due to 30 mins of continuous oscillation of the undulator in silicone oil (viscosity 0.97 Pa·s) at a constant V_w .

2. *Below equation (2), there is a line: “Two interesting observations ...”. It would be nice to explicitly mention the two observations as First and Secondly (or (i) and (ii))...*

Reply: We have modified the text as per the reviewer’s suggestion.

3. *It seemed to me that the present study was very much inspired by the previous observations by several authors of this paper on freshwater snail feeding. One may argue that this is a simplified experimental realization of that system. Yet the example of the snail is only abruptly mentioned in the introduction and not discussed later on. The authors should reconsider their phrasing and highlight their previous work appropriately.*

Reply: We agree that the example of snail was mentioned abruptly without giving a proper context. In the revised introduction we have made significant modifications which are as follows -

Liquid-air interfaces also act a cradle to Neuston, organisms that inhabit at and below the water surface. For example, the underwater apple snail *Pomacea canaliculata* exploits the water surface to drive a large scale surface flow and fetch floating food particles from afar in a process called *pedal surface collection* [24-27]. Whirligig beetles and water striders have developed morphological features to manipulate flow around them for effective locomotion at the interface [28,29]. Understanding the physics behind these natural phenomena could open up new bio-inspired strategies for flow actuation and sensing at interfaces.

...Our design, inspired in part by the capability of underwater snails in creating flow through undulations of its flexible foot [26,27], produces travelling waves on an undulator...

Reply to reviewer 4:

I have read the work of Pandey et al. with great interest. Their manuscript is well writing and illustrated, and overall, very thorough. The quality of the results and the completeness of the work is in line with what one can expect from a publication in nature communications. The work also constitutes a significant improvement in our understanding of free surface boundary driven flows in the low Reynolds limit. As such, I would thus recommend to accept this manuscript with minimal revisions.

Here are a few questions I have, which I hope will help improving the quality of the manuscript.

Reply: We thank the reviewer for supporting the manuscript for publication and for the comments, addressing which has certainly helped in improving the overall quality of the manuscript.

1. In Fig1, I am not sure why it is necessary to remove the data from 0 to -20mm. We should be able to see it and then the authors might explain why this data has to be discarded.

Reply: We agree that the argument behind removing the velocity data at the edge of the undulator is not well articulated in the manuscript. However, we would like to point out that the complete range of data is shown in Fig S2 b. For a given wave speed, this plot shows how the velocity along different trajectories are identical at large distances but exhibit very different magnitudes near the undulator, within a distance of 20 mm from the edge. The reason is that at the undulator edge, the flow field changes rapidly from bulk to thin-film flow. The sharp change in the boundary gives rise to local, secondary flow-fields that causes non-uniform and unsteady surface velocities that differ across different tracer trajectories. That is why we disregard the data within this transition region. In the revised Supplementary information we have clarified these points in section II:

The flow-field transitions from bulk to thin-film flow near the edge of the undulator due to the sharp change in boundary. As a result, local, secondary flow-fields develop which cause non-uniform and unsteady surface velocities that differ across different tracer trajectories. Size of this transition region varies with V_w and H . For the specific case shown in Fig. S2b, the dashed line marks the size of this zone which is 20 mm. Thus for all parameters, we disregard the trajectories within the transition regime in calculating surface velocity vs distance plots...

In the main manuscript, we add a brief clarification -

At the indulator edge, a sharp change in the boundary gives rise to a transition region in the flow-field where surface floaters exhibit non-uniform and unsteady motion (cf. fig. S2b in SI). Thus we disregard the first 20 mm of data to avoid edge effects

Figure 5: Surface fluid velocity at $x=-70$ mm as a function of wave speeds, V_w .

2. *How robust is the optimal vs. position. In other words, why taking -50mm. Likewise, the error bars are increasing as we move to higher speeds. Is there a reason why that might be the case.*

Reply: The velocity-distance curve for the highest wave speed, marked in dark blue remains below the curves for intermediate wave speeds for the complete range of x values. At larger distances, velocity magnitude drops very fast, making it harder to differentiate among multiple curves. In the above figure, we have reproduced the fig. 1f inset for $x=-70$ mm to answer both questions of the reviewer. Indeed, presence of the optimal speed is not limited to any particular distance, and the increasing error bars for higher V_w is not systematic, as shown in the above plot. We do agree that these points could have been explained better which hope to do with the following modification in the revised manuscript

Interestingly, we observe non-monotonic response in the surface fluid velocity; for any distance ($-20 \leq x \leq 120$) speed of fluid parcels initially increases with V_w , but subsequently drops down with further increase in V_w . Once $|\bar{V}|$ at a fixed location ($x = -50$ mm) is plotted against V_w (inset of fig. 1f), it becomes apparent that the maximum surface flow is achieved for an intermediate speed, $V_w \simeq 80$ mm/s

3. *In Fig2a, the interface is sharp, while the picture is a long exposure. This is odd, and I imagine that the picture has been cropped.*

Reply: The reviewer is correct that the long-exposure image shown in fig 2a is cropped. The liquid-air interface creates reflection of the illuminated particle tracks and of the undulator. To avoid confusion, we have cropped the image. To provide further clarity, an uncropped version of the image is shown in Fig. 6. Location of the free surface is estimated by identifying the plane of reflection.

Figure 6: The un-cropped long-exposure shot of the tracer particles in the thin film of liquid above the undulator. The reflection about the interface is cropped out in the image shown in fig. 2a of the main manuscript.

4. It might be best to show the all scene. It would be nice to compare the profiles of h_f obtained in the model to that found in experiment. Fig4 would be a good place to do so.

Reply: A precise measurement of the free surface profile is beyond of the scope of this work. The interfacial profiles shown throughout the main manuscript and supplementary information are only estimations obtained from the laser-illuminated particles located close to the liquid-air interface. Thus resolution of these measurements are not sufficient for a direct comparison of theoretical profiles of fig. S5a.

In any case, I was impressed by the ingenious nature of the work and the quality of the theoretical treatment. I want to reiterate that this work deserves publication in nature comm.

We thank the reviewer once more for the encouraging words on the manuscript. We hope to have addressed his/her concerns.

REVIEWERS' COMMENTS

Reviewer #1 (Remarks to the Author):

The authors have addressed sufficiently almost all of my remarks regarding the technical aspects of the manuscript, which has now improved a lot.

In point 6, the authors mention that Q_{max} should increase with H^2 and that "This trend is apparent in the red and green data sets of fig.3b, but the blue data does not. We think that the intermediate thickness corresponding to blue data falls in the transition regime (see the rescaled data of fig.4c) where the asymptotic results do not hold".

If I am not mistaken, both the blue and the orange points (the two largest thicknesses - out of four data sets in total) do not follow the scaling $Q_{max} \sim H^2$. Moreover, the range of Ca/Bo numbers in fig 4c is not that different when compared to the green data sets. I think that adding a remark about this in the manuscript will help the interested reader understand the limitations of the assumptions involved in the theoretical model.

Regarding the novelty of the work, in the abstract it is mentioned that "Our findings reveal a novel mode of pumping with less energy dissipation near a free surface compared to a rigid boundary". To me this sentence is unclear. Although the mode of pumping through a moving carpet is indeed new, energy dissipation will always be less near a free deformable surface compared to a rigid boundary (regardless of the way transport is caused), as surface stresses will be 0 and maximum, respectively.

Reviewer #2 (Remarks to the Author):

The restriction of the pumping condition or a clear definition of pumping seems necessary to claim pumping in this paper. I was interested in a 3D sketch of a flow to get a better concept. The authors studied the flow of 5 planes on the carpet and showed them in Fig. S3. These flow fields are helpful in understanding the similarity of a flow over the carpet. After seeing the broader view, I realized that pumping in this study is confusing in a general sense. When we talk about pumping, we usually mean creating a net flow or pressure from one end to the other. Otherwise, it just ends up mixing a

fluid. As I understand it, a net flow was created across the wave carpet, and in that sense, pumping was achieved. On the other hand, a counterflow was created outside the carpet. To claim pumping, there must be a net flow or pressure from one end to the other.

Reviewer #3 (Remarks to the Author):

I am happy with the response of the authors, the new experiments, and the expanded SI. I recommend this for publication.

Reply to reviewer 1:

The authors have addressed sufficiently almost all of my remarks regarding the technical aspects of the manuscript, which has now improved a lot. In point 6, the authors mention that Q_{max} should increase with H^2 and that “This trend is apparent in the red and green data sets of fig.3b, but the blue data does not. We think that the intermediate thickness corresponding to blue data falls in the transition regime (see the rescaled data of fig.4c) where the asymptotic results do not hold”. If I am not mistaken, both the blue and the orange points (the two largest thicknesses - out of four data sets in total) do not follow the scaling $Q_{max} H^2$. Moreover, the range of Ca/Bo numbers in fig 4c is not that different when compared to the green data sets. I think that adding a remark about this in the manuscript will help the interested reader understand the limitations of the assumptions involved in the theoretical model.

Reply: We agree with the reviewer that the flux values for the two larger thicknesses does not follow the thin-film prediction and consequently these data sets do not follow the scaling of Q_{max} . In the revised manuscript, we have added the following note in the ‘Optimal wave speed’ section.

Note that the blue and orange data sets in fig. 4c exhibit optimal flow rate at lower Ca/Bo than the above asymptotic prediction. This difference could be attributed to higher H/λ where the thin-film equation can not capture the exact flow field.

Regarding the novelty of the work, in the abstract it is mentioned that “Our findings reveal a novel mode of pumping with less energy dissipation near a free surface compared to a rigid boundary”. To me this sentence is unclear. Although the mode of pumping through a moving carpet is indeed new, energy dissipation will always be less near a free deformable surface compared to a rigid boundary (regardless of the way transport is caused), as surface stresses will be 0 and maximum, respectively.

Reply: We thank the reviewer for pointing out this unclear statement in the abstract. Our intent was to reiterate the benefit of a free surface in transporting liquid at lower amount of dissipation.

In the revised manuscript, we modify this sentence as: **Our findings reveal how proximity to free surfaces, which ensure lower energy dissipation, can be leveraged to achieve directional transport of liquids.**

Reply to reviewer 2:

The restriction of the pumping condition or a clear definition of pumping seems necessary to claim pumping in this paper. I was interested in a 3D sketch of a flow to get a better concept. The authors studied the flow of 5 planes on the carpet and showed them in Fig. S3. These flow fields are helpful in understanding the similarity of a flow over the carpet. After seeing the broader view, I realized that pumping in this study is confusing in a general sense. When we talk about pumping, we usually mean creating a net flow or pressure from one end to the other. Otherwise, it just ends up mixing a fluid. As I understand it, a net flow was created across the wave carpet, and in that sense, pumping was achieved. On the other hand, a counterflow was created outside the carpet. To claim pumping, there must be a net flow or pressure from one end to the other

Reply: We would like to emphasize that the flow over an undulating carpet is not enclosed within a channel in our system. Instead, we have the presence of the free surface, which does not allow us to impose a pressure gradient. The undulating boundary, in coordination with the free-surface, drives a net flow of liquid which we refer to as pumping. Since the oscillating carpet is placed inside an open tank, the liquid which comes out of the back of the carpet, gets pushed to the sides, creating a backflow that the reviewer refers to. This re-circulation is a consequence of the open nature of the set-up, rather than a limitation of our unique pumping mechanism. In the revised ‘Large scale flow’ section, we have spelled out these features -

We quantify pumping from the net flow that the undulator creates near the free surface. Unlike closed pipes and channels, this net flow is not driven by an imposed pressure gradient. The shape of the free surface, which is unknown a priori, determines the local pressure inside the liquid film. The undulator’s open ends and proximity to the tank walls result in flow re-circulation observed in fig. 1d.